# Effects of Alternative Cassava and Taro Feed on the Carcass and Meat Quality of Fattening Pigs Reared under Ecuadorian Backyard Systems

**DOI:** 10.3390/ani13193086

**Published:** 2023-10-03

**Authors:** Alfredo Valverde Lucio, Ana Gonzalez-Martínez, Julio Gabriel Ortega, Evangelina Rodero Serrano

**Affiliations:** 1Faculty of Natural Sciences and Agriculture, University of the South of Manabí (UNESUM), Jipijapa 130303, Manabí, Ecuador; yhonny.valverde@unesum.edu.ec (A.V.L.); julio.gabriel@unesum.edu.ec (J.G.O.); 2Department of Animal Production, Faculty of Veterinary Sciences, University of Cordoba, AGR-134, ceiA3., 14071 Córdoba, Spain; agmartinez@uco.es

**Keywords:** alternative diets, meat quality, morphometric traits, gastrointestinal tract

## Abstract

**Simple Summary:**

Pork is currently the cheapest protein source in the world. In the traditional rearing of backyard pigs in regions of Ecuador, cassava and taro crops are frequently used as replacement alternatives to corn in pig feed formulations. In this study, the quality and characteristics of the carcass and the behavior of the gastrointestinal tract (GIT) of 30 fattening pigs reared under the backyard production system were analyzed. The animals were fed with a conventional or alternative diet based on the addition of cassava and taro in doses of 32% and 42%. The results showed a higher effect of the geographical location than the feed administered to the animals. The morphological traits were those with lower changes between groups than the gastrointestinal tract measurements. The proportions of alternatives used in the formulations must be optimized, since this directly increases the amount of protein in the meat and the weight of the GIT, decreasing the degree of fattening of the carcass. In the production of backyard pigs in Ecuador based on the use of by-products and agricultural waste, it is necessary to promote the standardization of the type of pig that is raised, taking into account geographical location and promoting the use of local genetic resources.

**Abstract:**

Ecuadorian small producers use crossbred animals with a low level of genetic improvement, which are fed with alternative feeds to decrease production costs. The objective of this study was to evaluate the effects of geographical location and three diets according to the amount of cassava and taro incorporated into the feed (T1 conventional feed; T2 and T3 with 32% and 42% of cassava and taro, respectively) in pigs reared under the backyard system. The results did not show many differences between the treatments for morphological traits; however, between geographical locations, significant differences were evidenced. The fat content from the first rib was higher in the T1 group. The intramuscular fat percentage was higher in the T1 group, contrary to the protein levels, which were higher in the T3 group in Esmeraldas and the T2 group in Ro Chico. In the gastrointestinal tract (GIT) and its attached organs, differences were found in the empty stomach weight, full and empty small intestine weight, liver weight, and total GIT weight, with the T2 and T3 groups having the largest and heaviest. Cassava and taro did not affect the morphometric behavior and quality of the carcass but increased the amount of protein in the meat and the weight of the GIT. Geographical location was also observed to have a significant effect.

## 1. Introduction

Pork is currently the cheapest protein source in the world [1]. Its production reached 122 million tons in 2021, positioning it as the second-highest and the highest production and consumption worldwide, respectively [2]. In Ecuador, pork production in 2021 was 220,000 t, which was supported by backyard producers [3]. This backyard family production system traditionally uses agricultural feed alternatives generated from cultivation on their own farms, as well as cooking by-products to reduce production costs [4,5].

Small producers in Ecuador lack technical infrastructure as well as health plans. They use crossbred animals with a low level of genetic improvement, resulting from unplanned crossings between pure and crossbred animals, or between improved mixed breeds [6,7], according to the geographical location [5,8]; this directly affects the productivity, as it is a key factor for weight gain [9] and the quality of the carcass [10].

The quality of meat is defined by its palatability and consumer acceptance [11,12], and at the organoleptic level, it is measured by its color, smell, texture [13], and fat content, in addition to other technical aspects such as its pH, water retention capacity, and fatty acid and cholesterol profile [14]. It can also be defined by its health benefits (e.g., amounts of omega 3, vitamins, and amino acids) [15].

Production systems can interfere with obtaining a quality carcass [16,17]. In this sense, pigs raised in outdoor production systems, in which they consume pastures and complementary diets, grow healthier, and have better productivity [18], offer higher carcass yields and more tender meat, as well as a greater amount of intramuscular fat, unsaturated fatty acids, vitamin E, and antioxidants [19,20].

Pig production systems involve production costs in which feed accounts for at least 70% [21], with feed alternatives representing a sustainable and economical way of feeding [22], which, when dosed correctly, do not affect the quality of the pork [23].

The feed alternatives must be formulated considering the nutritional requirements of the animals to guarantee their productive performance [24], without disregarding the age of the animals, as well as the intestinal needs over a period of time between three and four weeks to adapt to the new diet [25].

The production of backyard pigs is represented by the social stratum [26] and constitutes an important source of income for the family economy, both as an accessible source of protein and as a tradable good in the market [27]. Currently, pig production and its derivatives are an important source of employment [28], which contributes to social development by guaranteeing food security [29] and supplying the needs of the population with quality meat [30].

The feed alternatives used in the breeding and fattening of backyard pigs in Ecuador include a diversity of feeds, among which cassava, taro, tagua, bananas, and squash stand out [5]. The use of cassava (*Manihot esculenta*) and taro (*Colocacia esculenta*) as corn substitutes lowers the production costs of backyard pigs [22]. In traditional diets, maize represents between 50% and 70% of the diet content, which considerably increases the costs of production [31]. The use of cassava and taro as a food alternative due to their high digestibility [32,33,34] provides acceptable results for production [12,22,35,36]. However, due to the content of antinutritional factors in both feed alternatives, it is necessary to subject them to prior cooking to reduce their negative effects [37,38,39,40].

There is no scientific evidence regarding the simultaneous use of cassava and taro in the quality of the carcass of backyard pigs. Therefore, the objective of this research was to evaluate the characteristics of the carcass, the quality of the meat, and the behavior of the gastrointestinal tract of fattening pigs fed with cassava and taro and raised under backyard production systems in Ecuador. Secondly, the effect of the amount of alternative feed in the diet was evaluated, as well as the geographical location of the animals, which is directly related to their genetic origin.

## 2. Materials and Methods

### 2.1. Selection and Preparation of Animals

A total of 42 castrated crossbred pigs (20 males and 22 females) were used, which were purchased at 60 days of age from producers in the study area. The animals used were Creole pigs mixed with the Pietrain breed coming from few litters to have greater homogeneity (two per geographical location). The experiments were carried out in two geographical locations (Figure 1), with the purpose of carrying out repetitions of the study. Fifteen pigs were raised in Quinindé, Esmeraldas province, and the remaining fifteen in Río Chico, Manabí province. Both geographical locations have a tropical climate in terms of their average annual temperatures, rainfall, and altitudes [22]. The experiments were conducted during July to November 2021.

The animals from each geographical location were randomly distributed into three groups (four males and three females per group), and they were housed in traditional pig pens [22] with an area of 1.25 m^2^ per animal. Before the experiment began, the pigs were given a ten-day period to adapt to the location, and a progressive change in their feed was carried out. It is worth noting that the volume of experimental feed was gradually increased every 5 days. The pigs were provided with water ad libitum through feeding bottles. Each group was given a different diet formulation (T1, T2, and T3, according to the following section). The feed was supplied twice a day at fixed times, at 8:00 a.m. and 3:00 p.m. Prior to the start of the experiment, the pigs had a period of ten days to adapt to the new feed, with the amount increasing progressively.

### 2.2. Preparation and Formulation of Diets

The feed alternative based on by-products of the cassava and taro processing industry were used. Both feed alternatives were administered to the animals after cooking to eliminate anti-nutritional components. Prior to cooking, the cassava and taro were weighed, washed, and chopped with the peel included; in addition, salt was added to increase the palatability of both feeds for the animals. After cooking, the feed was allowed to cool before mixing it with the rest of the components of the diet administered to the animals.

Three diets were formulated: one without the addition of cassava and taro (T1), and the remaining two with the addition of 32% (T2) and 42% (T3) of feed alternative in equal parts (Table 1). The protein and energy content of the diet was standardized according to the productive phase the pigs were in, which was growth or fattening. Thus, during the growth stage, the amount of protein was 18%, and in the fattening stage, it was 15%. The bromatological analysis of the feed alternatives used, as well as the formulas used for the experiment, were described in a previous study [22]. The animals received this diet during the 90 days that the trial lasted. All of the animals remained healthy throughout the experiment.

### 2.3. Procedure for Obtaining the Data

The pigs were utilized in the development of two studies; one was recently published [22]. For the second piece of research, only 30 pigs were utilized, first choosing all the males in each lot and then females until there were 5 animals per group.

The animals, at 160 days and 77.84 ± 1.71 kg, were slaughtered after a ten-hour fasting period, in accordance with Ecuadorian regulations [41]. Each carcass was weighed and measured while hot, and morphometric measurements of the foreleg, leg, ham, and shank were conducted. The rest of the measurements were obtained 24 h after slaughter [42]. The quartering of the carcass was carried out according to the indications of Nieto et al. [43]. The head was removed by cutting at the occipito-atlas joint, and the feet by cutting at the carpus-metacarpal and tarsus-metatarsal joints. The carcass was split longitudinally and, finally, to prevent dehydration, kept at −20 °C in plastic bags. After 24 h since the slaughter had passed, the loin was separated by a cut that began just ventral to the ventral side of the scapula at the cranial end and followed the natural curvature of the vertebral column to the ventral edge of the psoas major at the caudal end of the loin. The ham was removed with a straight cut between the second and third sacral vertebrae, and then the foreleg was separated from the trunk. After the rib was separated from the vertebrae, measurements of the fat were taken at the first and last rib levels as well as haunch point. Once all of the parts (head, loin, ham, foreleg, ribs, and legs) had been separated, they were weighed and measured.

The following measurements were collected: (i) the weights of the head, loin, ribs, ham, foreleg, and legs; (ii) the length of the carcass, bone, and muscle of the foreleg and ham; (iii) the perimeters of the front shank and ham; (iv) the thickness of backfat (DBT) at the first and last rib levels and gluteus point; and (v) the loin and haunch fat. The measurement instruments used were a high-precision digital scale from Montero, model TCS300JC61Z©, with a range of 300 kg to 2000 g (d = 100 g); a RexBeti Stainless Hardened © digital vernier caliper (measuring range: 5906 in. Precision: 0.1 inch); and a Jontex © brand digital scale with a maximum capacity of 40 kg and a minimum of 200 g (e = d = 5 g).

The digestive viscera of the gastrointestinal tract (GIT), stomach, liver, pancreas, small intestine, colon, cecum, and rectum were separated from the carcass to be individually measured and weighed, first full and then empty. The total weight of the viscera was calculated using the sum of the individual weights of each one of the parts, obtaining a weight for the total of the full GIT and another for the empty GIT [44].

The collection of weights and measures was carried out by the same technician for the two locations in order to reduce potential errors in obtaining the data [9].

For the meat quality analysis, a sample of 200 g of the longissimus lumbar muscle was taken at the level of the last rib 45 min after slaughter and was frozen at a temperature between −18 and −20 °C [42]. The bromatological analyses to determine the content of protein, fat, dry matter, moisture, ash, and pH were carried out in the Multianalityca S.A. laboratory (Quito—Ecuador) (certified SAE LEN 09-008). The reference methods of analysis were the following: moisture, Association of Official Agricultural Chemists, AOAC 925.10; crude protein, AOAC 2001.11; fat, AOAC 2003.06; ash, AOAC 923.03; and pH, Ecuadorian Technical Standard (NTE) INEN ISO 4316:2014m. Finally, the dry matter was estimated through the following calculation based on methods established by Maclean et al. [45]: dry matter = (initial weight − dry weight)/initial weight.

### 2.4. Statistical Analysis

IBM SPSS Statistics (version 26) software was used to perform the statistical analyses. All of the records were considered to be quantitative variables. After checking the normality and homogeneity of the variables, a mixed ANOVA with repeated measurements analysis was conducted. The statistical model included the fixed effects of treatment (T) and location (L) and their interaction (T × L). The repeated effect was location, and the subject of the repeated measurements was the animal nested within a group. When the fixed effects were significant, differences between the least squares means were assessed by a paired *t*-test at 5%. Moreover, Pearson correlations between carcass measurements were investigated in order to assess the relationship between the morphometric and compositional variables of the carcass and the GIT.

## 3. Results

### 3.1. Pig Carcass Morphology

Both of the effects considered (geographical location and feeding treatment) showed different results in terms of the morphological carcass characteristics (Table 2). The geographical location significantly affected (*p* < 0.05) most of the carcass characteristics of the backyard pigs in Ecuador, with the exceptions of the carcass yield, hot carcass weight, ham weight, rib weight, and leg weight. Meanwhile, the treatment only led to significant differences (*p* < 0.05) in ham perimeter, which showed the highest values in those pigs fed with conventional feed. However, the animals from Quinindé fed with 42% cassava and taro (T3_42%), as well as the animals from Río Chico that did not receive a feed alternative (T1_control), showed higher values for most of the parameters considered. The ham weight was significantly higher in pigs fattened in Quinindé with 42% of feed alternative. In general, the coefficients of variation were the lowest in Río Chico, and they showed different values between treatments.

### 3.2. Fat Thickness and Content of Pig Carcass

The backfat thickness at the first rib level was significantly (*p* < 0.05) higher in the animals that did not receive a feed alternative (Río Chico = 2.30 cm; Quinindé = 2.07 cm) (Table 3). Meanwhile, the backfat thickness at the last rib level and haunch fat were significantly (*p* < 0.05) higher in the animals from Quinindé (T1 = 2.34 cm and 1.51 cm; T2 = 1.70 cm and 1.45 cm; T3 = 1.82 cm and 1.41 cm, respectively).

### 3.3. Pork Quality Analysis

The bromatological characters showed significant differences (*p* < 0.05) for the geographical location effect, except for the percentage of intramuscular fat (Table 4). The moisture content, protein, and ash were higher in animals raised in Río Chico, while the pH and percentage of dry matter were higher in pigs raised in Quinindé. On the contrary, the diet that the animals received only significantly affected (*p* < 0.05) the pH of the meat, this being higher in pigs fed with 42% of alternative feeds (Quinindé = 5.80; Río Chico = 5.57).

### 3.4. Morphometry Characteristics of Gastrointestinal Tract and Visceral Organs

The pigs reared in Río Chico presented significantly (*p* < 0.05) higher values in almost all of the gastrointestinal tract (GIT) variables, with the exception of the full and empty small intestine weight and the total GIT weight (Table 5). Regarding the diet administered to the animals, it significantly affected (*p* < 0.05) the liver weight, empty stomach weight, full and empty cecum weight, and full GIT total weight, with higher values in pigs fed with 42% cassava and taro.

### 3.5. Relationship between Carcass Measurements and Morphometry of Pigs’ Gastrointestinal Tracts

Table 6 shows the correlations between the morphometric variables of the GIT and those of the carcass. The significant (*p* < 0.05) correlations found between the total GIT weight and the different parts of GIT weight were expected. The results reveal the negative and significant relationship (*p* < 0.01) between the amount of fat in the different parts and the development of the GIT, especially in the small intestine and colon.

## 4. Discussion

The present study investigated the effects of the simultaneous addition of cassava and taro to the feed of pigs and their effects on the carcass characteristics of backyard-raised pigs. They are reared under extensive traditional production systems in developed countries characterized by a low number of animals, which are generally fed with feed derived from the farmer’s own crops and kitchen waste; Creole or crossbred pigs are often used, and technological advances have been poor [5]. This study follows a previous study examining the effects of these same alternatives on growth and fattening parameters, in which it was possible to verify that the simultaneous use of both feed alternatives yields good productive results, in addition to lowering production costs by considerably reducing the amount of maize in the diet [22].

Differences in the carcass characteristics based on geographic location, as described by Schinckel and De Lange [46], were attributed to both changes in genetic selection and the environment in which the animals are reared. The tests carried out used crossbred pigs purchased from local producers, with only a small selection of animals highly specialized in meat production. Crossbreeding in the Ecuadorian backyard pig is very frequent, expressing very diverse phenotypes that vary from one producer to another [5,47]. Despite the fact that the choice of animals was random when forming the groups, and that the environmental and breeding conditions were similar, there are many differences between the locations, which suggests a genetic heterogeneity in the subjects that make up the sample; this corresponds to the reality of backyard pig farming systems in Ecuador, and the results of the treatments must be interpreted within the context of each of the two experimental locations. These differences were primarily found in weight and performance carcass parameters, fat thickness, and the development of the gastrointestinal tract. The differences in the coefficients of variation of the carcass yield between treatments could reflect variation in the live and carcass weights of each group, as well as the higher development of the gastrointestinal tracts in animals fed with the feed alternative.

Environmental temperature is an aspect to consider in pig farming because animals can suffer from thermal stress when raised in environments with temperatures above 25 °C [48]. A high temperature reduces feed consumption, affecting energy metabolism, increasing the accumulation of subcutaneous fat, which affects the quality of the meat [49]. This could be one of the reasons why the pigs fattened in Quinindé have high fat thickness, in contrast to the pigs from Río Chico that had heavier carcasses and meat pieces with higher yields, as well as greater development of the gastrointestinal tract. Despite the environmental similarities of the two geographical areas, in view of the commercialization of pork produced under backyard farming systems in Ecuador, it should be taken into account that the heterogeneity of crossbred pigs also gives rise to characteristic differences in their carcasses.

The genetic origin of the animals is another aspect to take into account, since backyard pig producers use crossbred pigs [5]. They come from crossing Creole pigs and foreign breed pigs, obtaining an increase in genetic variability, benefiting the pigs’ hardiness, immunological efficiency, and productive behavior [50]. In previous research, we studied backyard pigs in Ecuador and found that the breed most commonly used in this system was Creole, following by a crossbreed and Pietrain breed, although some farms reared a white pig breed such as Landrace [5]. However, other studies reveal that among the imported breeds preferentially used in this production system is the Duroc Jersey, since it is a dual-purpose breed, useful for meat and fat [51], followed by the Pietrain breed, whose characteristic is producing lean meat and little fat [52]. The results suggest the need to promote the breeding of standardized genetic models, for which native Creole-based local resources may be a good option; however, studies are needed to characterize the variability of these genetic resources [9,16].

The addition of 10% cassava leaf in the diet during the fattening phase of pigs has been found to improve the carcass characteristics in relation to conventional feed [1]. However, our results have revealed that backyard pigs from Ecuador fed with cassava and taro did not show differences in the morphological characteristics of the carcass. In addition, we did not find that different percentages of the alternative feed affected the size of the carcass cuts or the quality of the meat; thus, we consider that the most optimal formulation is T2, which includes 32% of the alternative feed. This formulation was also previously shown to be the most economically favorable in terms of productivity, without causing negative effects on the health of fattening pigs [22].

Gonzalez et al. [53] observed that cassava flour significantly improved body weight and had an impact on meat quality, with lower fat content being observed following treatments with cassava. However, comparing the results obtained when cassava was administered to the pigs in foliage and flour form at the same time shows that the results were similar to those obtained in our research, with higher carcass weights and yields, as well as the highest backfat thickness [54]. The addition of cassava with rice in the pig feed, replacing corn, produced the lowest carcass yields and lengths, as well as the lowest backfat thickness, although the differences between groups were not significant [55]. In the same way, the use of 40% fermented cassava in pig feed affected the fat content, moisture content, and ash in the carcass, as well as the protein content of the meat [56]. The last was also found in our results, as the meat derived from pigs fed with cassava and taro showed a higher protein content and a lower degree of fatness. Coinciding with our results, the addition of cassava or taro causes the animal to accumulate less fat, which is evidenced by a decrease in the thickness of the subcutaneous and intramuscular fat in pigs [1,57].

Hasan et al. [12] used cassava by-products (foliage, pulp, and peel) in proportions of 20, 40, and 60% in the feeding of weaned pigs. Their results determined that, both at a physical level (pH, color, and water retention capacity) and at a chemical level (protein and fat), the best treatment was the one that contained 20% cassava by-products.

Our results show that the addition of cassava and taro to the diet of pigs leads to an increase in organ weight, which is consistent with the results observed by Caicedo et al. [58] when testing the addition of different percentages of taro as a substitute for corn. Kaensombath and Lindberg [57] found similar results when soybean meal was replaced with ensiled taro leaves. One reason for the increase in organ weight in pigs fed with taro could be related to the ingestion of oxalate [57], but we cooked the alternative feed to avoid the presence of anti-nutritional factors such as oxalates. However, Taysayavong et al. [59] did not find differences in visceral organ weight and length in Moo Lath and Large White breeds, although they affirmed that their results could be explained by the short experimental period, which was only twelve days.

A greater consumption of feed motivates a greater development of the GIT, which justifies its greater development when providing moist feeds, since a greater volume of feed is given per moisture calculation [60]. The increase in the feed allowance leads to important increases in the weight of the total viscera, liver, kidneys, etc. [43]. For their part, Fitzsimons et al. [61] point out that, in general, the amount of energy provided in the food could influence the weight of the liver and the gastrointestinal tract. Coinciding with Ortega et al. [62], our results show that this leads to a decrease in the degree of fattening of the carcass in all parts. The addition of a large amount of fiber in the diet contributes to development of the GIT [57,63], which explains the results obtained in our study. Since the cassava and taro were administered whole, with the peel included, this provided extra fiber content to the diet of the animals [22].

## 5. Conclusions

The addition of cassava and taro residues as an alternative in the diet of pigs raised in the traditional backyard production system of Ecuador can be considered an alternative to reduce the use of corn without greatly affecting the morphological characteristics of the carcass. However, this affects their performance, as there is an increase in the weight of the gastrointestinal tracts of pigs during fattening. However, the environmental conditions and genetic origin could determine the geographical differences in these aspects.

The proportions of cassava and taro alternatives used in the formulations must be optimized since they directly increase the amount of protein in the meat and decrease the degree of fatness in the carcass.

In the production of backyard pigs in Ecuador based on the use of by-products and agricultural wastes, it is necessary to promote the standardization of the type of pig that is raised, promoting the use of local genetic resources.

## Figures and Tables

**Figure 1 animals-13-03086-f001:**
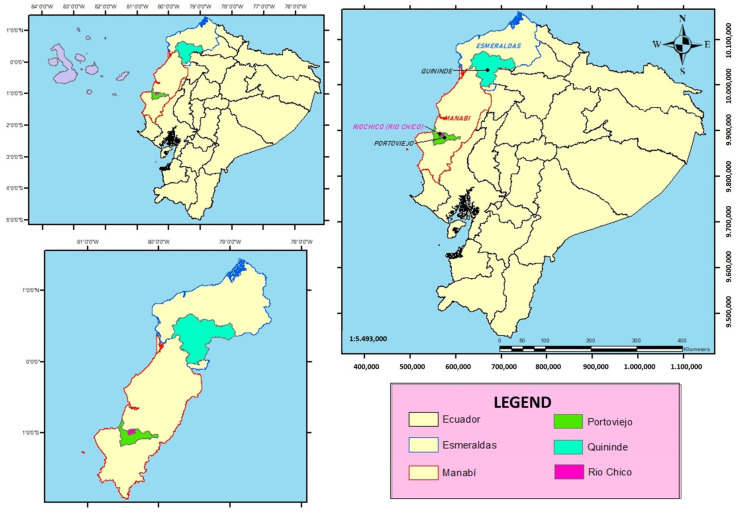
Geographic location of the sampling zones [22].

**Table 1 animals-13-03086-t001:** Nutritional value and composition of the used diets [22].

Ingredients ^1^	Phase					
	Growth			Fattening		
	T1 (Control)	T2 (32%)	T3 (42%)	T1 (Control)	T2 (32%)	T3 (42%)
Corn (kg)	23.64	9.55	4.55	23.64	9.55	6.36
Protein concentrate (kg) ^2^	13.18	16.36	17.27	11.36	15.00	16.36
Rice powder (kg)	8.18	4.55	4.09	10.00	5.91	3.18
Cooked cassava (kg)		7.27	9.55		7.27	9.55
Cooked taro (kg)		7.27	9.55		7.27	9.55
Red palm oil (kg)	1	1	1	1	1	1
Salt (g)	0.25	0.25	0.25	0.25	0.25	0.25
Crude protein (%)	17.35	17.35	17.35	15.30	15.29	15.29
Gross energy (kcal/kg)	3098	3079	3077	3084	3079	3094

^1^ Quantity per kilo of feed. ^2^ Soybean meal, rice by-products, banana meal, free fatty acids, molasses, calcium carbonate, mycotoxin binders, vitamin supplements (A, D3, E, and K3), riboflavin, niacin, thiamine mononitrate, cyanocobalamin, pyridoxine hydrochloride, biotin, trace mineral supplements, manganese sulfate, zinc sulfate, copper sulfate, ferrous sulfate, sodium, calcium iodide, methionine, lysine (such as hydrochloride and sulfate), threonine, choline chloride, antifungals, enzymes, antibiotics, and antioxidants.

**Table 2 animals-13-03086-t002:** Morphological characteristics of carcasses (mean ± standard error (coefficient of variation)) of backyard pigs fed with different formulations of nutritional alternatives with cassava and taro from two locations in Ecuador (Quinindé and Río Chico).

Traits ^1^	Quinindé			Río Chico			Location (L)	Treatment (T)	L × T
	Treatments ^2^								
	T1 (Control)	T2 (32%)	T3 (42%)	T1 (Control)	T2 (32%)	T3 (42%)			
LV (kg)	66.98 ± 5.36 (17.87) ^c^	73.37 ± 4.32 (13.16) ^bc^	77.83 ± 3.29 (9.45) ^abc^	87.91 ± 1.12 (2.85) ^a^	79.58 ± 1.62 (4.57) ^abc^	81.35 ± 1.27 (3.49) ^ab^	*p* < 0.01	0.200	0.051
HCW (kg)	47.50 ± 4.75 (21.44)	51.55 ± 2.95 (12.78)	55.87 ± 3.73 (14.93)	60.21 ± 0.61 (2.27)	55.12 ± 2.44 (9.91)	53.66 ± 1.75 (7.28)	0.087	0.902	0.086
CY (%)	0.71 ± 0.2 (6.38)	0.70 ± 0.01 (2.58)	0.72 ± 0.03 (8.36)	0.69 ± 0.01 (1.96)	0.69 ± 0.03 (9.55)	0.66 ± 0.02 (6.06)	0.083	0.867	0.506
CL (cm)	62.00 ± 0.77 (2.79) ^b^	63.00 ± 1.14 (4.05) ^b^	63.00 ± 1.14 (4.05) ^b^	75.72 ± 2.40 (7.15) ^a^	74.72 ± 1.31 (3.89) ^a^	73.72 ± 2.10 (6.37) ^a^	*p* < 0.01	0.943	0.670
FLL (cm)	29.00 ± 1.32 (9.90) ^a^	29.80 ± 0.20 (4.05) ^a^	31.20 ± 0.58 (4.18) ^a^	28.97 ± 0.47 (3.66) ^abc^	27.07 ± 0.37 (3.15) ^bc^	27.72 ± 0.44 (3.45) ^c^	*p* < 0.01	0.276	*p* < 0.01
LL (cm)	61.20 ± 1.71 (6.26) ^ab^	63.80 ± 1.32 (4.62) ^a^	63.00 ± 1.22 (4.35) ^ab^	57.6 ± 0.42 (1.64) ^bc^	54.96 ± 1.66 (6.76) ^c^	54.04 ± 0.99 (4.11) ^c^	*p* < 0.01	0.744	0.082
HL (cm)	43.60 ± 1.69 (8.67) ^a^	45.40 ± 0.81 (4.00) ^a^	43.40 ± 1.72 (8.86) ^a^	35.11 ± 0.54 (3.45) ^b^	33.58 ± 1.36 (9.05) ^b^	33.64 ± 0.99 (6.61) ^b^	*p* < 0.01	0.712	0.427
HP (cm)	75.40 ± 1.91 (5.67) ^a^	69.60 ± 1.81 (5.80) ^ab^	74.20 ± 1.69 (5.08) ^ab^	74.13 ± 1.26 (3.79) ^ab^	68.18 ± 2.42 (7.93) ^ab^	67.08 ± 1.67 (5.57) ^b^	*p* < 0.05	*p* < 0.05	0.209
FSP (cm)	14.60 ± 0.40 (6.13) ^b^	14.40 ± 0.51 (9.92) ^b^	14.80 ± 0.20 (3.02) ^b^	15.82 ± 0.27 (3.78) ^a^	15.28 ± 0.58 (8.48) ^a^	14.84 ± 0.30 (4.54) ^b^	*p* < 0.05	0.554	0.332
LW (kg)	5.46 ± 0.30 (12.48) ^bc^	5.26 ± 0.44 (18.86) ^c^	5.96 ± 0.40 (14.98) ^abc^	6.74 ± 0.06 (2.00) ^a^	6.65 ± 0.17 (5.61) ^ab^	6.68 ± 0.12 (4.11) ^ab^	*p* < 0.01	0.467	0.478
HW (kg)	13.89 ± 0.38 (6.15)	15.96 ± 1.5 (21.00)	17.34 ± 1.13 (14.59)	17.30 ± 0.31 (4.02)	15.54 ± 1.15 (16.53)	15.22 ± 0.70 (4.11)	0.712	0.760	*p* < 0.05
FW (kg)	10.66 ± 0.46 (9.64) ^ab^	8.80 ± 0.68 (17.38) ^b^	10.72 ± 0.61 (12.66) ^ab^	12.09 ± 0.28 (5.22) ^a^	11.25 ± 0.49 (9.80) ^a^	10.98 ± 0.57 (11.56) ^ab^	*p* < 0.01	0.054	0.141
RW (kg)	5.65 ± 0.35 (14.04)	4.86 ± 0.38 (17.46)	5.7 ± 0.6 (23.38)	5.66 ± 0.05 (1.91)	5.3 ± 0.22 (9.44)	5.18 ± 0.20 (8.82)	0.933	0.265	0.387
HDW (kg)	4.63 ± 0.10 (5.84) ^b^	4.64 ± 0.12 (5.87) ^b^	4.68 ± 0.29 (13.80) ^b^	6.18 ± 0.28 (10.08) ^a^	5.26 ± 0.50 (21.08) ^ab^	5.02 ± 0.43 (19.24) ^ab^	*p* < 0.01	0.201	0.162
FTW (kg)	1.11 ± 0.03 (5.84)	1.09 ± 0.06 (11.87)	1.11 ± 0.04 (8.06)	1.26 ± 0.07 (13.31)	1.16 ± 0.11 (21.49)	1.09 ± 0.06 (11.33)	0.262	0.425	0.472

^1^ LV: live weight; HCW: hot carcass weight; CY: carcass yield; CL: carcass length; FLL: hand length; LL: leg length; HL: ham length; HP: ham perimeter; FSP: front shank perimeter; LW: loin weight; HW: ham weight; FW: foreleg weight; RW: rib weight; HDW: head weight; FTW: feet weight. ^2^ T1: conventional feed with no corn replacement; T2: corn replacement with 32% of cassava + taro; and T3: corn replacement with 42% of cassava + taro. In addition, ^a, b, c^ are for each control; least square means without a common superscript differ significantly (*p* < 0.05) between groups.

**Table 3 animals-13-03086-t003:** Fat contents of the carcass (mean ± standard error (coefficient of variation)) of backyard pigs fed with different formulations of nutritional alternatives with cassava and taro from two locations in Ecuador (Quinindé and Río Chico).

Traits ^1^	Quinindé			Río Chico			Location (L)	Treatment (T)	L × T
	Treatments ^2^								
	T1 (Control)	T2 (32%)	T3 (42%)	T1 (Control)	T2 (32%)	T3 (42%)			
DBT1 (cm)	2.07 ± 0.24 (26.02)	1.23 ± 0.08 (15.21)	1.86 ± 0.23 (27.99)	2.30 ± 0.36 (34.71)	1.71 ± 0.23 (30.39)	1.81 ± 0.24 (29.89)	0.276	*p* < 0.05	0.569
DBT2 (cm)	2.34 ± 0.25 (23.46) ^a^	1.70 ± 0.81 (40.57) ^ab^	1.82 ± 0.23 (27.89) ^a^	0.91 ± 0.03 (6.32) ^bc^	0.88 ± 0.10 (24.66) ^bc^	0.86 ± 0.08 (20.66) ^c^	*p* < 0.01	0.199	0.267
DBT3 (cm)	1.52 ± 0.21 (31.13)	1.35 ± 0.29 (48.08)	1.54 ± 0.07 (9.66)	1.28 ± 0.14 (23.55)	1.16 ± 0.14 (27.47)	1.37 ± 0.13 (20.46)	0.176	0.494	0.980
LF (cm)	1.56 ± 0.26 (37.01)	1.26 ± 0.13 (22.51)	1.32 ± 0.06 (9.68)	0.94 ± 0.23 (55)	1.11 ± 0.3 (60.1)	1.09 ± 0.14 (29.42)	0.057	0.953	0.486
HF (cm)	1.51 ± 0.26 (38.45) ^a^	1.45 ± 0.26 (40.03) ^a^	1.41 ± 0.07 (10.43) ^ab^	1.30 ± 0.25 (25.49) ^ab^	0.73 ± 0.09 (25.9) ^b^	0.92 ± 0.22 (53.87) ^b^	*p* < 0.01	0.245	0.421

^1^ DBT1: backfat thickness at first rib level; DBT2: backfat thickness at last rib level; DBT3: buttock fat; LF: loin fat; HF: haunch fat. ^2^ T1: conventional feed with no corn replacement; T2: corn replacement with 32% of cassava + taro; and T3: corn replacement with 42% of cassava + taro. In addition, ^a, b, c^ are for each control; least square means without a common superscript differ significantly (*p* < 0.05) between groups.

**Table 4 animals-13-03086-t004:** Bromatological analysis of meat (mean ± standard error (coefficient of variation)) from backyard pigs fed with different formulations of nutritional alternatives with cassava and taro from two locations in Ecuador (Quinindé and Río Chico).

Traits ^1^	Quinindé			Río Chico			Location (L)	Treatments (T)	L × T
	Treatments ^2^								
	T1 (Control)	T2 (32%)	T3 (42%)	T1 (Control)	T2 (32%)	T3 (42%)			
H %	64.46 ± 2.74 (9.51) ^bc^	62.56 ± 2.94 (10.51) ^c^	69.34 ± 2.35 (7.57) ^abc^	73.8 ± 0.45 (1.36) ^a^	72.96 ± 0.38 (1.18) ^ab^	76.41 ± 1.86 (5.44) ^a^	*p* < 0.01	0.054	0.714
CP %	17.37 ± 0.81 (10.40) ^c^	18.11 ± 0.66 (8.11) ^c^	18.93 ± 0.22 (2.54) ^bc^	21.51 ± 0.63 (6.54) ^ab^	23.33 ± 0.77 (7.43) ^a^	19.23 ± 1.16 (13.45) ^bc^	*p* < 0.01	0.097	*p* < 0.01
IMF %	2.98 ± 0.60 (45.26)	2.71 ± 0.21 (17.7)	1.75 ± 0.11 (14.69)	3.13 ± 0.50 (35.75)	2.21 ± 0.66 (66.83)	2.88 ± 1.26 (97.73)	0.631	0.513	0.484
Ash %	0.86 ± 0.05 (12.12) ^b^	0.82 ± 0.04 (10.77) ^b^	0.88 ± 0.04 (10.88) ^b^	1.42 ± 0.07 (10.48) ^a^	1.50 ± 0.04 (5.76) ^a^	1.48 ± 0.07 (10.38) ^a^	*p* < 0.01	0.756	0.531
pH	5.76 ± 0.05 (1.93) ^ab^	5.61 ± 0.02 (0.98) ^ab^	5.80 ± 0.08 (2.92) ^a^	5.54 ± 0.04 (1.78) ^bc^	5.34 ± 0.04 (1.78) ^c^	5.57 ± 0.08 (3.06) ^abc^	*p* < 0.01	*p* < 0.01	0.924
DM %	34.8 ± 2.80 (18.00) ^ab^	37.44 ± 2.94 (17.57) ^a^	30.66 ± 2.35 (17.12) ^abc^	26.2 ± 0.45 (3.83) ^bc^	27.04 ± 0.38 (3.17) ^bc^	23.59 ± 1.86 (17.62) ^c^	*p* < 0.01	0.061	0.727

^1^ H: humidity; CP: crude protein; IMF: intramuscular fat; DM: dry matter. ^2^ T1: conventional feed with no corn replacement; T2: corn replacement with 32% of cassava + taro; and T3: corn replacement with 42% of cassava + taro. In addition, ^a, b, c^ are for each control; least square means without a common superscript differ significantly (*p* < 0.05) between groups.

**Table 5 animals-13-03086-t005:** Behaviors of the digestive tracts (mean ± standard error (coefficient of variation)) of backyard pigs fed with different formulations of nutritional alternatives with cassava and taro from two locations in Ecuador (Quinindé and Río Chico).

Traits ^1^	Quinindé			Río Chico			*p*		
	Treatments ^2^						Location (L)	Treatment (T)	L × T
	T1 (Control)	T2 (32%)	T3 (42%)	T1 (Control)	T2 (32%)	T3 (42%)			
SW (kg)	0.12 ± 0.01 (21.22) ^b^	0.14 ± 0.02 (32.97) ^ab^	0.15 ± 0.02 (26.07) ^ab^	0.22 ± 0.05 (46.47) ^ab^	0.28 ± 0.06 (49.08) ^a^	0.23 ± 0.02 (18.95) ^ab^	*p* < 0.01	0.523	0.608
LW (kg)	1.17 ± 0.03 (5.46) ^b^	1.29 ± 0.04 (6.3) ^b^	1.24 ± 0.04 (6.42) ^b^	1.41 ± 0.09 (14.46) ^b^	1.41 ± 0.10 (16.51) ^b^	1.76 ± 0.04 (4.6) ^a^	*p* < 0.01	*p* < 0.05	*p* < 0.05
PW (kg)	0.13 ± 0.6 (13.55) ^b^	0.12 ± 0.01 (24.59) ^b^	0.12 ± 0.01 (14.65) ^b^	0.13 ± 0.003 (6.54) ^b^	0.16 ± 0.01 (16.38) ^a^	0.16 ± 0.01 (12.17) ^a^	*p* < 0.01	0.420	0.089
FEWL (kg)	1.32 ± 0.34 (56.88) ^b^	2.08 ± 0.37 (39.99) ^a^	1.73 ± 0.22 (27.93) ^ab^	1.73 ± 0.03 (3.26) ^ab^	1.37 ± 0.16 (25.48) ^b^	1.65 ± 0.06 (8.51) ^ab^	*p* < 0.05	0.742	0.168
ESW (kg)	0.48 ± 0.01 (5.89) ^b^	0.53 ± 0.03 (13.35) ^b^	0.51 ± 0.03 (12.23) ^b^	0.79 ± 0.03 (8.61) ^a^	0.44 ± 0.07 (35.06) ^b^	0.85 ± 0.03 (6.72) ^a^	*p* < 0.01	*p* < 0.01	*p* < 0.01
FSIW (kg)	2.17 ± 0.36 (37.26)	2.79 ± 0.43 (34.19)	2.70 ± 0.28 (23.2)	2.01 ± 0.1 (11)	2.58 ± 0.09 (7.83)	2.98 ± 0.21 (15.77)	0.793	*p* < 0.05	0.423
ESIW (kg)	1.27 ± 0.06 (11.38)	1.37 ± 0.04 (6.64)	1.40 ± 0.07 (11.94)	1.23 ± 0.04 (7.82)	1.50 ± 0.12 (18.11)	1.56 ± 0.07 (9.82)	0.257	*p* < 0.05	0.433
FCEW kg)	0.46 ± 0.07 (35.18) ^c^	0.60 ± 0.08 (30.97) ^bc^	0.76 ± 0.04 (10.97) ^ab^	0.82 ± 0.05 (12.86) ^a^	0.62 ± 0.04 (15.34) ^ab^	0.68 ± 0.03 (8.37) ^ab^	*p* < 0.05	0.066	*p* < 0.05
ECEW (kg)	0.12 ± 0.01 (15.66) ^c^	0.13 ± 0.01 (13.28) ^bc^	0.15 ± 0.01 (12.09) ^ab^	0.15 ± 0.01 (10.54) ^ab^	0.17 ± 0.01 (13.15) ^a^	0.14 ± 0.1 (10.10) ^ab^	*p* < 0.01	0.11	*p* < 0.01
FPRW (kg)	0.23 ± 0.02 (22.48) ^c^	0.25 ± 0.01 (10.2) ^c^	0.25 ± 0.01 (5.98) ^bc^	0.39 ± 0.02 (12.30) ^a^	0.34 ± 0.03 (17.52) ^ab^	0.38 ± 0.03 (15.58) ^a^	*p* < 0.01	0.719	0.228
EPRW (kg)	0.19 ± 0.01 (15.8) ^b^	0.22 ± 0.01 (6.48) ^b^	0.21 ± 0.01 (12.66) ^b^	0.32 ± 0.02 (16.98) ^a^	0.25 ± 0.02 (21.43) ^ab^	0.25 ± 0.02 (13.81) ^ab^	*p* < 0.01	0.318	*p* < 0.05
FCOWL (kg)	2.28 ± 0.27 (26.84) ^b^	2.62 ± 0.07 (6.07) ^ab^	2.00 ± 0.10 (10.68) ^b^	3.09 ± 0.12 (8.96) ^a^	3.12 ± 0.2 (14.47) ^a^	3.30 ± 0.16 (10.77) ^a^	*p* < 0.01	0.348	0.077
ECOW (kg)	0.87 ± 0.03 (8.35) ^c^	0.94 ± 0.04 (10.25) ^bc^	0.91 ± 0.02 (4.76) ^bc^	1.19 ± 0.12 (23.13) ^abc^	1.38 ± 0.18 (29.46) ^a^	1.34 ± 0.09 (15.53) ^ab^	*p* < 0.01	0.494	0.504
TWTF (kg)	6.43 ± 0.85 (29.67) ^c^	8.37 ± 0.61 (16.3) ^bc^	7.51 ± 0.50 (14.92) ^c^	9.93 ± 0.08 (1.83) ^b^	10.57 ± 0.15 (3.1) ^ab^	12.20 ± 0.47 (8.67) ^a^	*p* < 0.01	*p* < 0.01	0.074
TWTE (kg)	2.93 ± 0.10 (7.38) ^c^	3.19 ± 0.06 (4.28) ^bc^	3.16 ± 0.10 (7.34) ^bc^	3.67± 0.16 (9.66) ^ab^	3.74 ± 0.21 (12.73) ^ab^	4.13 ± 0.12 (6.69) ^a^	0.068	*p* < 0.01	0.095

^1^ SW: spleen weight; LW: liver weight; PW: pancreas weight; FEWL: full stomach weight; ESW: empty stomach weight; FSIW: full small intestine weight; ESIW: empty small intestine weight; FCEW: full cecum weight; ECEW: empty cecum weight; FPRW: full pig rectum weight; EPRW: empty pig rectum weight; FCOWL: full colon weight; ECOW: empty colon weight; TWTF: full total gastrointestinal tract weight; TWTE: empty total gastrointestinal tract weight. ^2^ T1: conventional feed with no corn replacement; T2: corn replacement with 32% of cassava + taro; and T3: corn replacement with 42% of cassava + taro. In addition, ^a, b, c^ are for each control; least square means without a common superscript differ significantly (*p* < 0.05) between groups.

**Table 6 animals-13-03086-t006:** Pearson correlation coefficients between carcass characters and those of the gastrointestinal tract.

Traits ^1^	HW (kg)	RW (kg)	DBT1 (cm)	LF (cm)	HF (cm)	P (%)	IMF (%)	LW (kg)	ESIW (kg)	ECEW (kg)	ECOW (kg)	TWTE (kg)
HCW (kg)	0.77 **	0.85 **	0.35	−0.01	0.06	0.20	−0.02	0.19	0.03	0.13	−0.01	0.16
HW (kg)		0.48 *	0.16	0.02	0.19	0.24	0.00	0.19	0.03	0.09	−0.05	0.15
RW (kg)			0.29	0.12	0.04	−0.01	−0.04	0.02	−0.09	0.01	−0.17	−0.02
DBT1 (cm)				−0.17	−0.06	−0.07	0.19	0.31	−0.01	0.18	−0.03	0.19
LF (cm)					−0.41 *	−0.13	−0.16	−0.30	−0.09	−0.24	−0.42 *	−0.27
HF (cm)						−0.38 *	−0.19	−0.40 *	−0.5 **	−0.13	−0.39 *	−0.42 *
P (%)							−0.02	0.36	0.58 **	0.15	0.43 *	0.50 **
IMF (%)								0.135	0.01	−0.09	−0.01	0.04
LW (kg)									0.74 **	0.78 **	0.48 **	0.92 **
ESIW (kg)										0.56 **	0.65 **	0.89 **
ECEW (kg)											0.41 *	0.76 **
ECOW (kg)												0.68 **

^1^ HCW: hot carcass weight; HW: ham weight; RW: rib weight; DBT1: backfat thickness at first rib level; LF: loin fat; HF: haunch fat; P: protein; IMF: intramuscular fat; LW: live weight; ESIW: empty small intestine weight; ECEW: empty cecum weight; ECOW: empty colon weight; TWTE: empty total gastrointestinal tract weight. * *p* < 0.05; ** *p* < 0.01.

## Data Availability

This is not applicable, as the data are not in any data repository with public access. However, if an editorial committee needs access, we will happily provide them with it. Please use this email: erodero@uco.es.

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
