# Peer review of "Effects of Alternative Cassava and Taro Feed on the Carcass and Meat Quality of Fattening Pigs Reared under Ecuadorian Backyard Systems"

_animals, 2023, doi:10.3390/ani13193086_

Round 1

Reviewer 1 Report

Dear authors

I appreciate the relevance of the topic of the article. The use of alternative sources of animal nutrition and ensuring the high nutritional quality of the produced meat is a highly recommended topic of these days. Nevertheless, in the future, it is advisable to adjust the breeding methodology, especially the division of animals based on gender. Gender and feed conversion are closely related and the given results could alternatively be biased. It is clear to me that in small farms there is no room for dividing animals based on sex. In the future, it would be appropriate to compare individual genders separately.

Nevertheless, I recommend your article for publication.

Quality of english is sufficient. The reader can clearly understand the objectives and the results of the research.  

Author Response

Dear,

We would like to thank for your work and valuable comments that have substantially helped us to improve the former manuscript quality. All your comments have been considered and corrections have been made according to them. To make your work easier, we have highlighted the changes you and rest of reviewers requested by using track changes function. Also, the manuscript has been revised by a MDPI Author Services to improve the English.

We hope that you like the new version of the manuscript.

Thank you very much for your interest.

Kind Regards,

The authors

Reviewer 2 Report

The manuscript is well written, but covers a rather narrow area in which the analyzed components could be used.

Minor comments:

Please present the recipe composition of feed mixtures and their chemical composition not only by referring to publications but also in tabular form with a reference to the work in which this data is included.

Please use international abbreviations for: moisture; proteins;

There is no need to present the Pvalue  to four decimal places.

Author Response

(The authors gave the same response as above.)

Reviewer 3 Report

The manuscript "Effects of Alternative Cassava and Taro Feed on the Carcass and Meat Quality of Fattening Pigs reared under Ecuadorian Backyard Systems" is interesting and presents a novel and alternative feed for pig diets that affects some meat and carcass traits. However, there are some questions about the methodology and presentation of obtained results and discussion.

Material and methods

The methodology should be better described. It is not acceptable in this form. I suggest the author to show the geographical location of the experiment on the map (as Figure 1). They have also used the „backyard“ term but have not explained what type of system it is.

The authors did not write how many animals were kept per group. The results obtained are also questionable. You really need to explain this.

L95 – specifies which crosbreds. Is the number of F and M animals  the same (?), this needs to be better explained.

L100-L101 – rewrite the sentence

L102 – how many M/F per group?

L103 – italicise „ad libitum“

L109-L110 – rewrite the sentence

L110-L112 – this sentence should be deleted or moved to the discussion section

How much cassava and taro did you add to the diet? It is also compulsory to write down the composition of diets.

L127 – Authors must provide some important information, such as the age and final weight of the pigs.

L129-L130 – this needs to be rewritten. It needs to be better explained what meat parameters and carcass characteristics you measured after slaughter. The dissection procedure also needs to be revised.

RESULTS

The results should be better presented. I would recommend the authors to improve this section with clearer sentences.

Table 1. Please explain the high variation of CY in all tretmans. Also delete the „p“ in the table, it is clear that 0.01 refers to the p-values. This comment refers to all tables in the manuscript.

L171-L172 – please rewrite this sentence.

L173 – „The first of them“, please name the characteristic; rewrite „most of the characteristics of the carcasses“ to "most of the carcass characteristics“

L176 – which parameter?

L179 – delete numerically

L180-L181 – this sentence is unclear, please rephrase

L191 – please write a sentence explaining what is shown in table 2

L192 – does this refer to backfat thickness? Please explain this sentence

L241 - please rephrase this sentence

L243 – delete „or bacon“

L206 – delete considered

L207 – does this refer to all gruops or between groups? This is unclear and would need to be explained.

L208 – add the % for the numbers

L206-L213 – the authors are comparing the number that are not in Table 2. It is confusing, please check the results. Please also delete a dot in Table 2 for the 76..41

L222 - add a sentence reproducing the information in Table 3

DISCUSSION

The discussion is very poor and scarce. The results obtained are not compared with other studies and if they are, the factors studied are not coincident (different animals; ducks,…). The discussion needs to be improved!

 L253 – „In this research, for the first time, the effects of the simultaneous addition to feed of cassava and taro on the characteristics of the carcass of backyard-raised pigs was evaluated.“ Rewrite: „The present study investigated the effects of simultaneous addition of cassava and taro to the feed of pigs and their effects on carcass characteristics of backyard rised pigs.“ I would also recommend explaining the backyard farming system. Does this refer to extensive farming? Please specify.

L259 – rewrite „ characteristics of the carcass“ as„carcass characteristics“

L261- this „found“ is unclear. Does it refers on reared?

L261-L263 – „The tests carried out used crossbred pigs purchased from local producers, with only a small selection of animals highly specialised in meat production.“

L264- Replace yield with expressing.

The authors need to explain the difference between geographical locations so that this parameter can be considered as one of the effects in the study. The genetic basis of the pigs is also unclear.

The authors argue the influence of high temperatures on back fat thickness, but we do not know the genetic background of these pigs and whether there is a difference in fat accumulation between the crosbreds studied.

L290 – delete the dot (.)

L284 – L287 – this sentence is too long and needs to be reworded. It is unclear whether producers use only Duroc or Pietrain for crosses with the native Creole pig breed Why do they use Pietrain when we know that Duroc is often used for crosses with native pig breeds?

L295 - rewrite „ characteristics of the carcass“ as „carcass characteristics“

L 296 – delete this part of the sentence „and when comparing conventional feeds with those to which kitchen waste has been added [53].“

L297 – delete „conventional feed“

L300 – „carcass pieces“ on whitch traits does this refer? Please specify.

L304 – L307 – this refers to a study on duck. Please delete this. The authors should explain their results and compare them with the results of other authors whose research was done on pigs.

L320-L323 – Delete this sentence

L324-L327 – Please explain this better. I suggest rewriting the entire paragraph, explaining Kaensombath and Lindberg's study and comparing it to your results. I suggest rewriting the sentence as follows: „Our results show that the addition of cassava and taro to the diet of pigs leads to an increase in organ weight, which is consistent with the results observed by Caicedo et al [58]„.

CONCLUSION

This section is well written.

The English needs to be improved. Some sentences are too long and difficult to understand.

Author Response

(The authors gave the same response as above.)

Round 2

Reviewer 3 Report

Dear Authors,

thank you for Your responses and corrections. 

Kind regards